# IMAGEDIT: LET ANY SUBJECT TRANSFORM

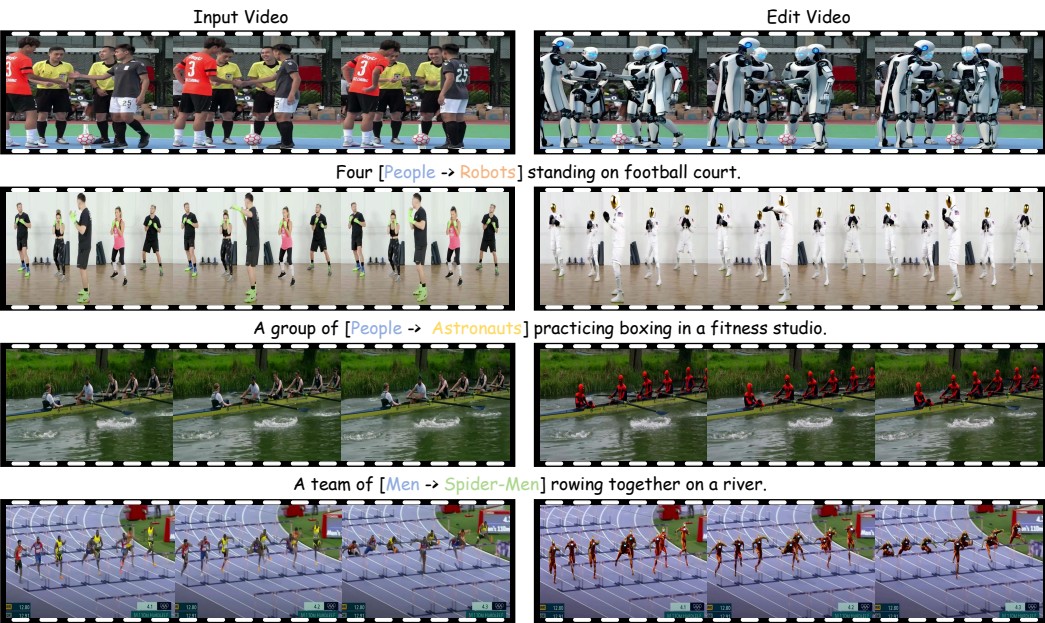

Figure 1: **Visualization results of IMAGEdit.** Given any video with any number of designated subjects, IMAGEdit performs precise category transformations while maintaining subject count and spatial layout. Especially in crowded scenes with overlapping subjects, IMAGEdit demonstrates stable consistent editing.

## ABSTRACT

In this paper, we present **IMAGEdit**, a training-free framework for any number of video subject editing that manipulates the appearances of multiple designated subjects while preserving non-target regions, without finetuning or retraining. We achieve this by providing robust multimodal conditioning and precise mask sequences through a prompt-guided multimodal alignment module and a prior-based mask retargeting module. We first leverage large models' understanding and generation capabilities to produce multimodal information and mask motion sequences for multiple subjects across various types. Then, the obtained prior mask sequences are fed into a pretrained mask-driven video generation model to synthesize the edited video. With strong generalization capability, IMAGEdit remedies insufficient prompt-side multimodal conditioning and overcomes mask boundary entanglement in videos with any number of subjects, thereby significantly expanding the applicability of video editing. More importantly, IMAGEdit is compatible with any mask-driven video generation model, significantly improving overall performance. Extensive experiments on our newly constructed multi-subject benchmark MSVBench verify that IMAGEdit consistently surpasses state-of-the-art methods. Code, dataset, and weights will be released.

## 1 INTRODUCTION

"Any subjects can transform together." -Many people voiced this wish as children while watching films, animations, and live performances. Television media often have such applications (Shen et al.,

2025), e.g., the coordinated team transformation in *Ultraman*[1] and the multi subjects synchronized transformation in *Sailor Moon*[2]. Reproducing this effect in real videos typically requires specialized equipment and extensive character modeling, increasing cost and limiting generalization. In this work, to *let any subject transform* while preserving non-target regions, we propose a novel, training-free framework for video editing with any number of subjects. As shown in Figure 1, even in scenes with any number of subjects where spatial relations are complex and interactions are dense, conditions that differ markedly from single or few subject settings of existing methods (Wu et al., 2023; Ceylan et al., 2023), our framework performs the edits reliably and achieves remarkable results.

With the rapid progress of generative models, video editing (Wu et al., 2023; Ceylan et al., 2023) has advanced substantially, driven by generative adversarial networks (Radford et al., 2015; Goodfellow et al., 2020; Donahue et al., 2016; Odena et al., 2017) and diffusion models (Rombach et al., 2022; Ramesh et al., 2022; Shen et al., 2025). However, most existing approaches (Geyer et al., 2023; Wang et al., 2025; Ceylan et al., 2023; Ku et al., 2024) focus on single or at most two subjects and typically rely on either task-specific training or precise guiding masks, which limits their generalization ability. For instance, as seen in the first row of Figure 2, although existing methods can achieve accurate editing in terms of position and quantity with precise masks, the subject categories do not faithfully reflect the editing prompt, highlighting limitations in the edited conditions. In multi-subject scenarios with dense layouts and heavy occlusions, these methods often become unstable, degrading perceptual quality. As shown in the second row of Figure 2, boundary entanglement in segmentation (Ren et al., 2016; He et al., 2017) can cause edits to spill across subjects, misplacing attributes, such as a dog head on a robot wolf body. Due to limited compositional grounding of prompts and control conditions, attention is diluted across subjects, leading to temporal inconsistency and disrupting edit continuity. In summary, editing videos with many subjects is more challenging than single or few subject cases. Occlusion and boundary entanglement make segmentation, tracking, and identity preservation error-prone, while instructions and control conditions must be accurately grounded to multiple subjects to avoid attention dispersion and ensure consistent edits and temporal coherence.

To address these limitations, we propose **IMAGEdit**, a training-free video editing framework that transforms any number of subjects in arbitrary videos without additional training. As shown in Figure 1, IMAGEdit delivers robust and precise edits across subjects of any number and is particularly effective in cases with boundary entanglement. This is achieved through three components: (i) a prompt-guided multimodal alignment module, (ii) a prior-based mask retargeting module, and (iii) a mask-driven video generation model.

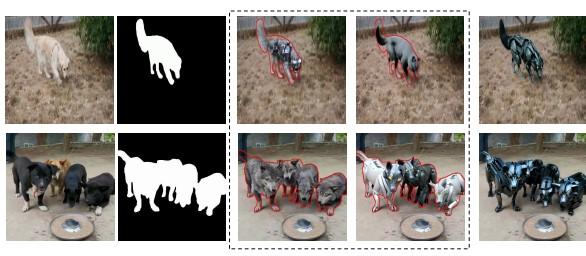

Reference Frame   Mask Frame          Existing Methods          IMAGEdit

Figure 2: Visual results generated from current video editing methods and our IMAGEdit (Dogs → Robot Wolves). Previous methods apparently retain the reference dog's appearance. In contrast, the result of IMAGEdit both aligns the robot wolf's features and captures the reference dog's layout.

In our prompt-guided multimodal alignment module, we first extract the subjects to be edited from the prompt and input a pretrained text-to-image (T2I) model (Rombach et al., 2022; Podell et al., 2023; Chen et al., 2023) to obtain the target appearance. We then feed both the editing prompt and the visual prior into a vision language model (VLM) (Achiam et al., 2023; Wang et al., 2024; Chen et al., 2024) to produce aligned multimodal conditions, namely an expanded text condition and an expanded image condition. For the second component, we present a theoretical algorithm to capture per-frame mask state changes and generate a temporal continuous mask motion sequence aligned with the input video. Finally, for the third component, we input the multimodal conditions and the continuous mask sequence into a pretrained mask-driven video generation model to transform the video. From Figure 2, IMAGEdit achieves reliable and coherent video edits by supplying multimodal conditions and retarget masks. Moreover, IMAGEdit operates as a plug in and is compatible

---

[1] https://en.wikipedia.org/wiki/Ultraman_(manga)#Anime
[2] https://en.wikipedia.org/wiki/Sailor_Moon#Live-action_film_&_series

with any mask-driven video generator, markedly improving multi subject editing performance, with experimental analysis in Sec. 4.2.

In addition, to address the lack of a benchmark for editing videos with any number of subjects, we construct MSVBench, which comprises 100 cases covering diverse subject counts and scene complexities. Qualitative and quantitative evaluations on MSVBench show that IMAGEdit delivers strong video editing performance and surpasses existing methods particularly in multi subject settings. Ablation studies further validate the effectiveness and advantages of the framework, offering valuable insights for the community. We also release IMAGEdit results on multi subject videos, providing a practical solution for research and applications in video editing.

Our main contributions are summarized as follows:

- We propose IMAGEdit, a novel training free video editing framework that enables the transformation of any number of subjects in arbitrary videos.
- IMAGEdit generates robust multimodal conditions and precise mask sequences for any number of subjects, offering a promising solution to the community for video editing.
- IMAGEdit can be seamlessly integrated as a plug in with any mask driven video generation model, consistently enhancing its performance in multi subject scenarios.
- We establish MSVBench, a benchmark with varying subjects for comprehensive evaluation. Experiments on MSVBench show that IMAGEdit outperforms SOTA approaches.

## 2 RELATED WORK

**Video editing.** Early video editing methods mainly relied on GANs (Goodfellow et al., 2020; Mittal et al., 2017; Pan et al., 2017; Li et al., 2018), performing subject edits through warping and rendering pipelines. In recent years, latent diffusion models( (Rombach et al., 2022; Peebles & Xie, 2023; Ruiz et al., 2023))have markedly improved the quality and efficiency of image generation. Building on this progress, several works fine-tune T2I models (Wu et al., 2023; Qi et al., 2023; Liu et al., 2024a; Zhang et al., 2025) with spatiotemporal attention on paired samples from a single video to achieve stylization and subject replacement. However, current one-shot training tends to overfit the given sample and fails to align with other target scenes; the issue is exacerbated in unseen, multi-subject, high-density settings, thereby limiting the generalization ability. Meanwhile, another line of research (Wang et al., 2025; Yang et al., 2025; Jiang et al., 2025; Bian et al., 2025) leverages highly scalable conditions, such as instance segmentation masks, to strengthen spatial localization and motion constraints. Yet, this approach inherently depends on masks and is restricted in multi-subject scenarios with overlapping and intertwined instances. To overcome these limitations and truly *let any subject transform*, this paper adopts a mask-driven video editing paradigm that provides precise retargeted mask sequences, enabling high-fidelity and robust any subject video editing.

**Instant Segmentation.** Instance segmentation aims to produce pixel-level masks for all objects in an image while distinguishing individual instances. Early approaches (Rother et al., 2004) constructed masks for region proposals and refined them iteratively to match instance extents. With the rise of deep networks, one line of work (Ren et al., 2016; He et al., 2017; Li et al., 2017) performs direct regression of instance masks using coarse-to-fine cascade networks. At the same time, another (Zhang et al., 2021; Cheng et al., 2021; 2022)predicts per-instance mask heatmaps or query embeddings for indirect regression, improving accuracy. Recently, prompted segmentation (Kirillov et al., 2023; Ravi et al., 2024; Ren et al., 2024) has been introduced with larger datasets and foundational models to enhance cross-domain generalization. Nevertheless, these methods (Rother et al., 2004; Ren et al., 2016) still struggle in dense scenes due to the supervised training paradigm and annotation constraints, particularly with many subjects. To this end, we adopt a prior-based mask retargeting module that exploits spatial semantic correspondences in deep features and strong generalization, providing precise and temporal consistent instance masks for any number of subjects.

**Text to Video Generation.** In recent years, image-to-video (I2V) generation (Singer et al., 2022; Yang et al., 2024b) has attracted considerable attention due to its potential in image animation and video synthesis. Prior work (Guo et al., 2023) leveraging diffusion models' strong representation and synthesis capabilities for image inserts temporal layers into pretrained two-dimensional U-Nets (Ronneberger et al., 2015) and fine-tunes with video data to convert static images into dy-

Figure 3: The IMAGEdit framework first derives robust multimodal cues via a prompt-guided multimodal alignment. Then, a prior-based mask retargeting module produces a time-consistent mask sequence aligned with the input video. Finally, the multimodal cues and mask sequence are fed into a video generation model to synthesize the edited video.

namic sequences. For example, VideoPainter (Bian et al., 2025) is a dual-branch framework that integrates with video diffusion transformers to achieve robust arbitrary-mask video inpainting. In parallel, specialized I2V frameworks (Singer et al., 2022; Ho et al., 2022) trained from scratch on large-scale, high-quality datasets have demonstrated strong competitiveness. DiT-based I2V approaches (Hong et al., 2022; Yang et al., 2024b; Wan et al., 2025; Gao et al., 2025)) have recently become increasingly popular for their improved global coherence and controllability. Guided by these considerations, we adopt Wan2.1 (Wan et al., 2025) as the base I2V model in this work.

## 3 METHOD

The overall framework of IMAGEdit is shown in Figure 3. We first introduce the diffusion transformer basics in Sec. 3.1, followed by a description of the three core components in Sec. 3.2: prompt-guided multimodal alignment, prior-based mask retargeting, and the video generation model.

### 3.1 PRELIMINARIES

In IMAGEdit, we adopt Wan2.1 (Wan et al., 2025) as the base model for mask-guided matching, which comprises a variational autoencoder (Kingma & Welling, 2013), a umT5 text encoder (Chung et al., 2023), and a denoising diffusion transformer (DiT) (Peebles & Xie, 2023). While DiT variants have shown strong performance in image synthesis, DiT based pipelines for video editing remain relatively underexplored compared with UNet based counterparts, particularly in multi subject, mask conditioned settings. Unlike approaches (Geyer et al., 2023; Qi et al., 2023; Yatim et al., 2024) that rely on UNet (Ronneberger et al., 2015), DiT uses a Transformer backbone to model the diffusion process and to capture long range dependencies and global context. Let $x_0 \in \mathbb{R}^{H \times W \times C}$ denote a clean image with height $H$, width $W$, and channels $C$. The forward diffusion process gradually corrupts $x_0$ into $\{x_t\}_{t=1}^{T}$ over $T$ discrete steps by adding independent Gaussian noise $z_t \sim \mathcal{N}(0, I)$, where $I$ represents the identity matrix:

$$x_t = \sqrt{\alpha_t}\, x_{t-1} + \sqrt{1 - \alpha_t}\, z_t, \quad t = 1, \ldots, T, \tag{1}$$

where $\alpha_t \in (0, 1)$ is the variance preserving noise schedule at step $t$. The reverse diffusion process iteratively removes noise to recover $x_{t-1}$ from $x_t$. We model this step with $p_\theta(x_{t-1} \mid x_t)$, which represents the conditional probability distribution of the less noisy image $x_{t-1}$ given the more noisy image $x_t$:

$$p_\theta(x_{t-1} \mid x_t) = \mathcal{N}\big(x_{t-1}; \mu_\theta(x_t, t), \Sigma_\theta(x_t, t)\big), \tag{2}$$

where $\mu_\theta(x_t, t)$ and $\Sigma_\theta(x_t, t)$ are the mean and covariance predicted by the DiT with parameters $\theta$.

### 3.2 IMAGEdit: LET ANY SUBJECT TRANSFORM

Reviewing the results in Figure 2, we observe remarkable variance in editing performance across different subject counts and boundary complexities. A robust mask generation mechanism that can handle multiple interacting subjects is essential for achieving high-fidelity video editing. Prior approaches either rely on supervised segmentation models trained on annotated data, expand dataset diversity to improve generalization, or introduce new regularization terms to enhance mask consistency. However, under a supervised training paradigm, these methods still struggle to generalize to unseen categories and densely entangled multi-subject scenarios, often leading to boundary

entanglement and temporal instability. To address this, we propose a prompt-guided multimodal alignment module that combines textual and visual priors to generate robust editing conditions. In addition, we introduce a prior-based mask retargeting module that produces temporal consistent mask motion sequences across frames. Finally, a mask-driven video generation model is employed to synthesize high-fidelity and robust multi-subject video edits.

**Prompt-Guided Multimodal Alignment.** Recent studies (Yin et al., 2023; Singer et al., 2022) show that limited understanding ability of text encoders in video editing models often causes inconsistencies between editing results and the intended semantics when using naive text prompts. In multi subject editing scenarios, this issue becomes more pronounced. As shown in Figure 4 (a) top row, neighboring subjects dilute attention, and a naive prompt fails to impose a clear constraint on "astronaut," thus not triggering the intended edit. Another case is shown in Figure 4 (a) bottom row, where insufficient textual

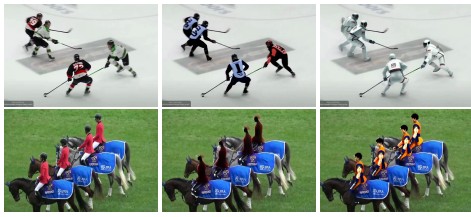

Figure 4: Visualization of the without (w/o) and with (w/) multimodal condition. The first row: Hockey Players → Astronauts; the second row: Horse Riders → Gokus.

semantic constraints cause a semantic mismatch, making "Goku", related attributes only partially take effect on the target. These observations indicate that multi-subject settings require stronger multimodal alignment and subject-level control to ensure precise binding of editing intent and temporal stability. Based on these observations, we introduce a prompt-guided multimodal alignment module to explicitly realize cross-modal alignment and produce stable multimodal conditions.

Specifically, as shown in Figure 5, we first extract subject specific tokens $W_{\text{ref}}$ from the original editing prompt $P_{\text{edit}}$. These tokens query a pretrained text to image model (Podell et al., 2023) to generate a visual prior $I_{\text{ref}}$, which bridges the abstract textual description and a concrete visual instance, anchoring the subject's appearance. Next, we feed $I_{\text{ref}}$ and $P_{\text{edit}}$ into a vision language model (VLM) (Achiam et al., 2023; Wang et al., 2024). Using an extended instruction template $P_{\text{temp}}$, the VLM aligns the two modalities by

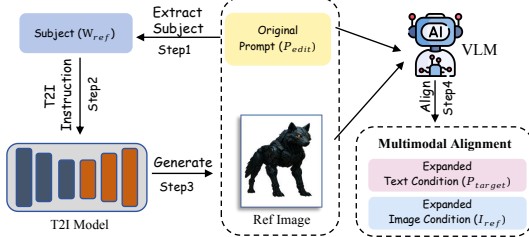

Figure 5: Illustration of prompt-guided multimodal alignment. We generate aligned extended text conditions and extended image conditions for each original prompt.

interpreting the visual attributes in $I_{\text{ref}}$ and expanding the description in $P_{\text{edit}}$ in a controlled manner. This yields an enriched and visually grounded textual condition $P_{\text{target}}$:

$$P_{\text{target}} = \Phi_{\text{VLM}}(P_{\text{edit}}, I_{\text{ref}} \mid P_{\text{temp}}), \tag{3}$$

where $\Phi_{\text{VLM}}$ denotes the VLM function that reconciles the semantic intent in $P_{\text{edit}}$ with the structure and appearance priors provided by $I_{\text{ref}}$. As shown in Figure 4 (b), grounding the textual expansion in explicit visual evidence improves the fidelity of subject attributes and mitigates attention dilution and semantic drift, resulting in more coherent and targeted video edits.

**Prior-Based Mask Retargeting.** The accuracy of masks directly determines the controllability and temporal stability of mask-driven video editing. In dense multi subject scenes, general segmentation models such as the SAM family often fail to produce precise instance level masks that distinguish overlapping or adjacent objects, and they cannot capture the hierarchical and occlusion order among subjects, leading to mask leakage and blurred boundaries; these errors further propagate and amplify over time, as shown in Figure 6 (a). To address this, we propose a prior-driven mask retargeting module: constrained by depth priors, it spatially reestimates instance boundaries according to near-far relationships and temporal generates a retargeted mask motion se-

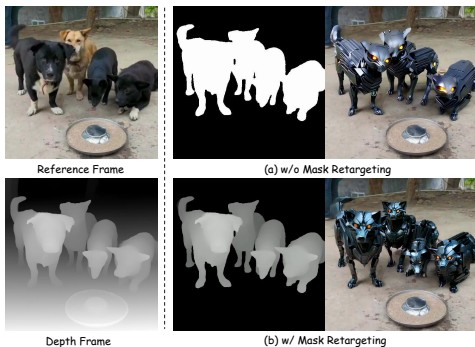

Figure 6: Visualization of the without (w/o) and with (w/) mask retargeting (Dogs → Robot Wolves).

quence by enforcing consistency across adjacent frames. This sequence explicitly encodes hierarchical boundaries and occlusion relationships between subjects, significantly reducing mask leakage and improving cross-frame consistency, as shown in Figure 6 (b).

As shown in Figure 3, we consider an original video $V_{\text{ori}} = \{v_1, v_2, \ldots, v_N\}$ with $N$ frames, where $v_i \in \mathbb{R}^{H \times W \times C}$. We denote the binary instance masks across frames by $M = \{m_1, m_2, \ldots, m_N\}$ with $m_i \in \{0, 1\}^{H \times W}$ specifying the editing region for frame $i$. Similarly, let $D = \{d_1, d_2, \ldots, d_N\}$ with $d_i \in \mathbb{R}^{H \times W \times C}$ denote the estimated depth maps. From these inputs, we extract guidance features using a conditional DiT. To obtain the mask-guided features $F^{\text{mask}}$, we first compute a masked video via element-wise multiplication with the binary mask: $V_{\text{masked}} = V_{\text{ori}} \odot M$. Each masked frame from $V_{\text{masked}}$ is then concatenated with its corresponding binary mask $m_i$ along the channel dimension and fed into the conditional DiT. The resulting output sequence is defined as $F^{\text{mask}} = \{F_i^{\text{mask}}\}_{i=1}^n$. Similarly, to get the depth-guided features $F^{\text{depth}}$, each depth map $d_i$ is concatenated with an all-ones mask and processed by a similar DiT architecture, yielding $F^{\text{depth}} = \{F_i^{\text{depth}}\}_{i=1}^n$. Subsequently, we achieve precise redirection of the mask region by injecting depth features $F^{\text{depth}}$ into the editing area. To ensure the depth information is fully integrated into the target area, avoiding missing or discontinuous information during feature fusion, and to provide a smooth transition for the fusion of depth features and mask features mask in the conditional module, we apply morphological dilation to the initial editing mask to expand the editing region. Formally, for each frame mask $m_i$, the dilated mask is

$$m_i'[p, q] = \max_{(u,v) \in \mathcal{N}_k} m_i[p + u, q + v], \tag{4}$$

where $\mathcal{N}_k = \{(u, v) | -r \leq u, v \leq r\}$ is a square neighborhood of size $k \times k$ with radius $r = \lfloor k/2 \rfloor$. This dilation enlarges the foreground to provide a blending margin. We then apply a Gaussian filter to $m_i'$ and downsample the result to obtain a soft mask $\tilde{m}_i$. Collectively, the final softened and resized mask sequence is $\tilde{M} = \{\tilde{m}_1, \tilde{m}_2, \ldots, \tilde{m}_N\}$. Let the final motion guidance sequence be $F^{\text{motion}} = \{F_i^{\text{motion}}\}_{i=1}^n$. At each spatial location $(x, y)$ we compute

$$F_i^{\text{motion}}(x, y) = \tilde{m}_i(x, y) F_i^{\text{depth}}(x, y) + (1 - \tilde{m}_i(x, y)) F_i^{\text{mask}}(x, y). \tag{5}$$

This design ensures that within subject regions (where $\tilde{m}_i$ is high), editing is primarily guided by depth to recover geometry and proper layering, while in background regions (where $\tilde{m}_i$ is low), mask constraints dominate to preserve appearance and temporal stability.

**Video Generation Model.** Given the retargeted mask motion sequence $F^{\text{motion}}$, we can condition any mask-driven video generator by attaching a ControlNet style branch to a ViT backbone (Wan et al., 2025; Gao et al., 2025; Zhang et al., 2023; Jiang et al., 2025). Although $F^{\text{motion}}$ can in principle be injected at all denoising steps, continuing the fusion at late steps produces severe artifacts and unnatural seams, because early steps shape low frequency structure while late steps refine high frequency details (Wu et al., 2024; Qian et al., 2024). We therefore inject only in the early structural phase and revert to mask only conditioning for refinement. Let $T$ be the total number of steps and $\tau$ the injection threshold. With depth guided features $F_{i,t}^{\text{depth}}$ and mask guided features $F_{i,t}^{\text{mask}}$, the conditional feature is

$$F_{i,t}^{\text{cond}}(x, y) = \begin{cases} \tilde{m}_i(x, y)\, F_{i,t}^{\text{depth}}(x, y) + \left(1 - \tilde{m}_i(x, y)\right) F_{i,t}^{\text{mask}}(x, y), & t \leq \tau, \\ F_{i,t}^{\text{mask}}(x, y), & t > \tau. \end{cases} \tag{6}$$

This scheme accurately tracks the motion encoded by the mask sequence, preserves high quality details, and generalizes across architectures, yielding consistent gains in multi subject scenarios.

## 4 EXPERIMENTS

**Datasets.** To comprehensively evaluate the effectiveness of multi-subject video editing methods in complex scenarios, we construct MSVBench. In this benchmark, over 60% of videos contain three or more subjects. It consists of 100 videos collected from YouTube [3] and TikTok [4], covering diverse subjects such as humans, animals, and vehicles, and intentionally includes multi-subject cases that

---

[3] https://www.youtube.com
[4] https://www.tiktok.com

are underrepresented in existing datasets. The number of subjects per video ranges from one to more than ten. For captions and editing prompts, we employ GPT-4o (Achiam et al., 2023) to generate scene descriptions for each video and automatically produce corresponding editing instructions based on subject attributes. All generated descriptions and prompts are manually verified to ensure accuracy and usability. Further details are provided in Appendix A.

**Metrics.** Following (Cong et al., 2023; Yang et al., 2025), we evaluate video editing fidelity using four metrics. Specifically, Warp-Err quantifies background consistency in non-edited regions. CLIP-T measures the alignment between the edited text and the edited regions; CLIP-F assesses perceptual consistency between adjacent frames. Moreover, Q-Edit is a composite indicator that reflects text alignment and temporal consistency. In addition, to assess spatial consistency before and after editing under varying numbers of subjects, we present center matching error (CM-Err), which performs one-to-one matching of subject boxes detected by GroundingDINO (Liu et al., 2024b) before and after editing and computes the mean center displacement. More details are provided in Appendix B.

**Implementation Details.** All experiments are conducted on a single NVIDIA A800 80 GB GPU. Unless stated otherwise, the configuration is as follows: (i) the denoising DiT and the conditional DiT are initialized from the pre-trained Wan2.1 (Jiang et al., 2025); (ii) the text to image model is the pre trained SDXL (Podell et al., 2023), and the vision language model is Qwen2.5 VL 32B Instruct (Bai et al., 2023); (iii) instance masks are obtained using Grounded SAM 2 (Ren et al., 2024), and depth maps are estimated with Depth Anything V2 (Yang et al., 2024a); (iv) at inference we use 50 denoising steps and set the injection threshold to $\tau = 30$.

## 4.1 MAIN RESULTS

We compare our proposed IMAGEdit with several state-of-the-art methods, including open-source approaches such as FateZero (Qi et al., 2023), TokenFlow (Geyer et al., 2023), VideoPainter (Bian et al., 2025), VideoGrain (Yang et al., 2025), and DMT (Yatim et al., 2024), as well as closed-source approaches such as Keling[5], Runway[6], and Viggle[7].

**Quantitative Results.** On MSVBench, the proposed IMAGEdit delivers consistently superior performance across all key metrics from Table 1. Concretely, it achieves the best scores with Warp-Err (1.85), CLIP-T (27.23), CLIP-F (97.93), Q-Edit (14.72), and CM-Err (2.83). Compared with the strongest open-source methods, IMAGEdit improves Q-Edit from 13.13 (DMT) to 14.72 (+12.1%), evidencing the benefit of robust multimodal features from the prompt-guided multimodal alignment for edit fidelity.

Table 1: Quantitative results on MSVBench comparing IMAGEdit with SOTA methods. The best score is in **bold**; the second-best is underlined. A superscript * denotes closed-source methods.

| Methods | Warp-Err ↓ | CLIP-T ↑ | CLIP-F ↑ | Q-Edit ↑ | CM-Err ↓ |
|---|---|---|---|---|---|
| FateZero | 2.16 | 24.26 | 97.42 | 11.23 | 3.49 |
| TokenFlow | 2.10 | 26.57 | 96.93 | 12.65 | 3.78 |
| VideoPainter | 2.05 | 24.13 | 96.97 | 11.77 | 4.70 |
| VideoGrain | 1.98 | 24.71 | 97.13 | 12.48 | 3.12 |
| DMT | 1.87 | 24.55 | 96.76 | 13.13 | 3.80 |
| Keling* | 2.00 | 25.66 | 98.37 | 12.83 | 4.39 |
| Runway* | 1.87 | 25.82 | 97.73 | 13.81 | 3.36 |
| Viggle* | 1.86 | 25.04 | 97.53 | 13.46 | 3.43 |
| **IMAGEdit** | **1.85** | **27.23** | **97.93** | **14.72** | **2.83** |

Meanwhile, it reduces CM-Err from 3.12 (VideoGrain) to 2.83 while slightly lowering Warp-Err from 1.87 (DMT) to 1.85, demonstrating that the precise masks produced by the prior-based mask retargeting improve temporal consistency and background preservation. Even against closed-source methods, IMAGEdit remains competitive, e.g., Q-Edit 14.72 vs. 13.81 (Runway). Overall, these results substantiate the effectiveness of our proposed IMAGEdit.

**Qualitative Results.** From Figure 7, most competing methods (e.g., FateZero, TokenFlow, and VideoGrain) suffer from boundary entanglement and attention dilution, often leading to incomplete edits, background corruption, or attribute leakage across subjects. In contrast, IMAGEdit correctly transforms the designated subjects while preserving non-target regions, indicating that the prompt-guided multimodal alignment supplies robust multimodal conditions that precisely drive subject conversion. Moreover, existing approaches exhibit poor temporal stability under limb motions and occlusions, methods such as VideoPainter and DMT frequently show missing subjects or reduced fidelity, whereas our prior-based mask retargeting produces consistent mask sequences, enabling

---

[5]`https://klingai.com/global`
[6]`https://runwayml.com`
[7]`https://viggle.ai`

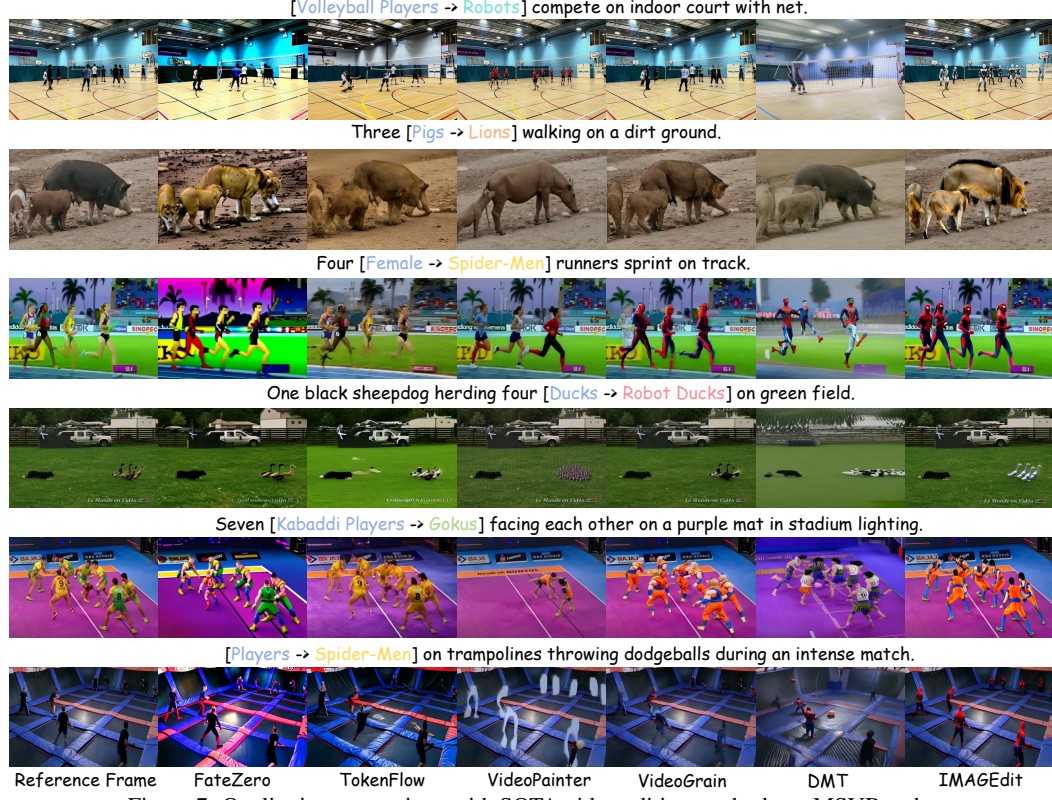

Figure 7: Qualitative comparison with SOTA video editing methods on MSVBench.

IMAGEdit to maintain frame-to-frame coherence and high fidelity under complex motion. Overall, our method yields more consistent and realistic edits, demonstrating the advantages of IMAGEdit.

**User Study.** The obtained quantitative and qualitative results underscore the substantial superiority of our IMAGEdit in generating results. To further validate the superiority of our method in human perception, we randomly selected 20 cases and recruited 20 volunteers to assess each method across three critical dimensions: Background Preservation (BP), Text Alignment (TA), and Video Quality (VQ). The volunteers ranked the edited videos according to these criteria to ensure a fair and comprehensive comparison across methods. As shown in Figure 8, IMAGEdit achieved the highest scores in BP, TA, and VQ, demonstrating its strong editing capability on videos with varying numbers of subjects.

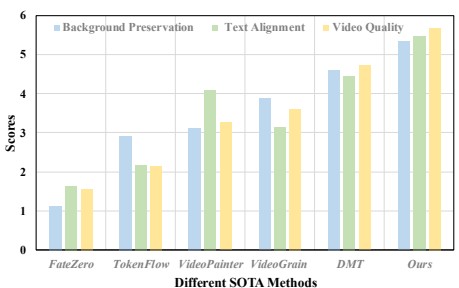

Figure 8: User study results. Higher values in these three metrics indicate better performance.

### 4.2 ABLATION STUDY

To assess the effectiveness of each component, we construct the following variants within the IMAGEdit framework, keeping all other settings fixed while altering component configurations: **B0**: the base video generation model only Wan2.1. **B1**: only the prior-based mask retargeting module enabled. **B2**: only the prompt-guided multimodal alignment module enabled.

**Prompt-Guided Multimodal Alignment.** As shown in Table 2, adding the prompt-guided multimodal alignment module (B2) already improves performance over the base model (B0), increasing CLIP-T from 24.78 to 26.12 and Q-Edit from 13.24 to 14.04, while reducing CM-

Table 2: Quantitative ablation results.

| Methods | CLIP-T ↑ | Q-Edit ↑ | CM-Err ↓ |
|---|---|---|---|
| B0 | 24.78 | 13.24 | 3.00 |
| B1 | 25.10 | 13.42 | 2.87 |
| B2 | 26.12 | 14.04 | 2.99 |
| **IMAGEdit** | **27.23** | **14.72** | **2.83** |

Err from 3.00 to 2.99. These improvements demonstrate that explicit alignment between textual prompts and visual priors provides stronger multimodal conditioning, leading to better adherence to

editing instructions and more consistent layouts. Visual comparisons are presented in Figure 9, it confirms that this module mitigates incomplete edits and attribute leakage, producing more accurate transformations across multiple subjects.

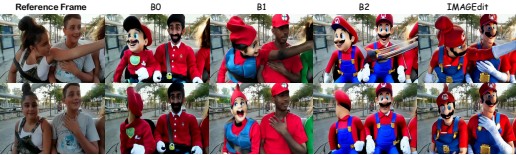

**Prior-Based Mask Retargeting.** From Table 2, incorporating the prior-based mask retargeting module (B1) yields clear improvements over the base model (B0), raising CLIP-T from 24.78 to 25.10 and Q-Edit from 13.24 to 13.42. Although CM-Err remains comparable, the generated masks are more precise and

Figure 9: Visualization of ablation results of IMAGEdit. (People → Super Mario)

temporal consistent, enabling edits to better follow the target subjects across frames. Figure 9 further shows that this module effectively reduces boundary entanglement and preserves non-target regions, leading to more stable and faithful video edits, particularly in multi-subject scenarios with dense interactions or occlusions.

### 4.3 MORE RESULTS

**Attention Weight Distribution.** As shown in Figure 10, we systematically evaluated the impact of prompt-guided multimodal alignment on the spatial distribution of cross-attention weights.

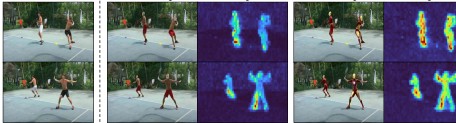

The target prompt is "Two Iron-Men are playing tennis on a tennis court." We visualized the cross-attention of "Iron-Men" to assess the weight distribution. Without prompt-guided multimodal alignment, the attention weight for "Iron-Men" appears only in certain areas, such as the head, leading to incomplete editing. In contrast, IMAGEdit evenly distributes the attention weight for "Iron-Men" across

Figure 10: Attention weight distribution for both without (w/o) and with (w/) multimodal condition. (Players → Iron-Men)

the entire body, which is correct. This is because prompt-guided multimodal alignment provides multimodal conditional information, allowing for better capture of the regions that need editing.

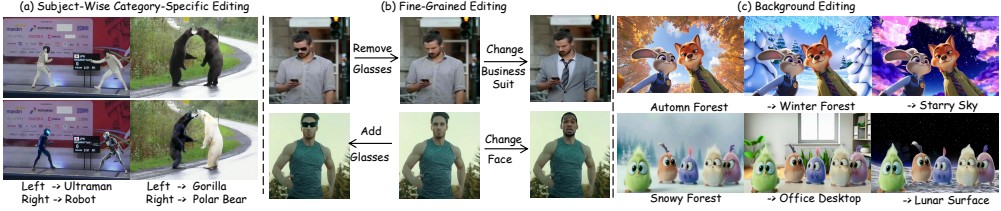

Figure 11: Results across multiple scenarios, demonstrating the extensibility of IMAGEdit.

**Multi-Scenario Applications.** IMAGEdit also performs strongly across diverse application scenarios, including subject wise category specific editing, fine grained editing, and background editing. Specifically, as shown in Figure 11 (a), we convert the left person into an ultraman and the right person into a robot; Figure 11 (b) demonstrates fine grained edits such as adding glasses and changing clothing; Figure 11 (c) edits the background to Autumn Forest, snowy forest, and starry sky styles. Overall, IMAGEdit maintains stable appearance and clean boundaries, indicating good scalability of the framework. Additional results are provided in Appendix C.

## 5 CONCLUSION

We presented IMAGEdit, a training free framework for video editing with any number of subjects that changes designated categories. IMAGEdit provides robust multimodal conditioning and precise mask motion sequences through two key components, a prompt guided multimodal alignment module and a prior based mask retargeting module. By leveraging the understanding and generation capabilities of large pretrained models, these components produce aligned multimodal signals and time consistent masks that effectively remedy insufficient prompt side conditioning and overcome mask boundary entanglement in crowded scenes. The framework then conditions a pretrained mask driven video generator to synthesize the edited video. IMAGEdit is plug and play with a wide range of mask driven backbones and consistently improves overall performance. Extensive experiments on the new multi subject benchmark MSVBench verify that IMAGEdit surpasses state of the art methods. Code, dataset, and weights will be released to support further research.

## 6 ETHICS STATEMENT

We are committed to conducting research in an ethical manner and ensuring that our work adheres to the highest standards of integrity. This research adheres to the ethical guidelines set forth by our institution and follows appropriate protocols for data collection, processing, and dissemination. Regarding the use of generative models and video editing technologies, we ensure that the content generated using IMAGEdit and similar tools does not infringe upon the rights of individuals or communities. We take steps to mitigate the potential for misuse of the technology, such as the creation of misleading or harmful content, and we emphasize the importance of responsible usage. Additionally, we acknowledge the potential societal impacts of advanced generative technologies. While our work aims to enhance creative and productive applications, we remain cognizant of the broader ethical concerns, such as privacy, consent, and the potential for deepfakes. We advocate for the responsible deployment of these technologies, with safeguards in place to protect individuals' rights and dignity. Lastly, all datasets used in our research are either publicly available, ensuring that data usage complies with privacy regulations and ethical standards.

## 7 REPRODUCIBILITY STATEMENT

To ensure the reproducibility of our work, we will make all relevant resources publicly available, including the code, datasets, and model weights. The code used for implementing IMAGEdit, along with the model weights, will be released on GitHub. Additionally, we will provide access to the datasets used in our experiments, either by directly sharing them or by linking to publicly available sources. This will enable others to reproduce our results and build upon our work.

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

## SUPPLEMENTARY MATERIAL

This supplementary material provides extended details for the methodology and experiments presented in the main paper. Section A details the MSVBench dataset. Section B describes the computation of the CM-Err metric. Section C reports additional results, including evaluations on extra datasets, broader qualitative comparisons, and further examples of scalable applications. Section D discusses potential avenues for future research. Section E documents the usage of large language models. More video results are available on: imag2025.github.io/imagedit/ [8].

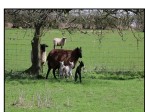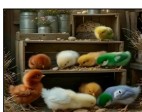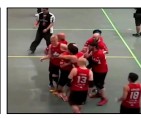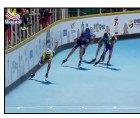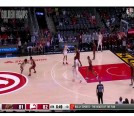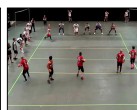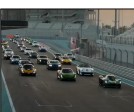

Figure 12: Display of randomly selected samples from the MSVBench dataset.

## A    MSVBENCH DATASET

To fill the evaluation gap in multi subject video editing, we construct MSVBench with 100 videos, more than sixty percent of which contain three or more subjects, as shown in Figure 12. Videos are primarily sourced from YouTube and TikTok. Scenes cover humans, animals, and vehicles; the number of subjects per frame ranges from one to more than ten, and the dataset includes challenging cases with crowded layouts, strong occlusions and interactions, significant camera motion, and complex backgrounds. Unlike prior editing datasets that focus on single subject or face centered clips, MSVBench is explicitly sampled and annotated around high subject count, dense layouts, and interaction or occlusion ordering, making it

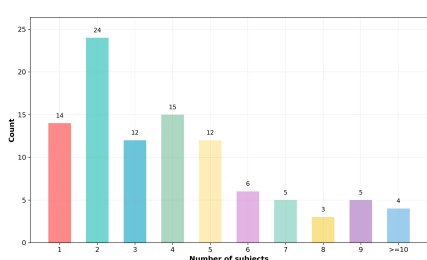

Figure 13: Distribution of the number of subjects in a video in MSVBench.

well suited to evaluate category fidelity, layout preservation, and boundary leakage. For annotation, GPT-4o (Achiam et al., 2023) generates concise video level descriptions and corresponding editing prompts, which are then verified by human annotators for accuracy and consistency; Grounded SAM2 (Ren et al., 2024) produces instance level masks for the target regions, followed by manual checks to ensure temporal consistency. We will release the verified descriptions and prompts, mask sequences, and evaluation scripts, and we report the distribution of subject counts in Figure 13 to facilitate reproduction and comparison.

## B    CENTER MATCHING ERROR METRIC

We assess subject count and layout consistency before and after editing with a layout aware, alignment free metric, since pixel overlap measures such as mIoU cannot capture merges, splits, or relocations. We introduce center matching error (CM-Err). For frame $t$ of width $W$ and height $H$, let $\mathcal{A}_t = \{a_j\}$ and $\mathcal{B}_t = \{b_k\}$ be the sets of bounding boxes from the original and edited frames. For a box $b = (x_{\min}, y_{\min}, x_{\max}, y_{\max})$, its center is $c(b) = \big((x_{\min} + x_{\max})/2, \, (y_{\min} + y_{\max})/2\big)$. The normalized center distance between $a_j$ and $b_k$ is

$$d_{jk}^{(t)} \; = \; \frac{\big\| c(a_j) - c(b_k) \big\|_2}{\sqrt{W^2 + H^2}} \; \in [0, 1]. \tag{7}$$

Using $d_{jk}^{(t)}$ as the cost, we compute the minimal one to one matching between $\mathcal{A}_t$ and $\mathcal{B}_t$; let $M_t$ be the number of matched pairs and $U_t = |\mathcal{A}_t| + |\mathcal{B}_t| - 2M_t$ the number of unmatched boxes. The frame level error is

$$\text{CM-Err}^{(t)} \; = \; \frac{\sum_{i=1}^{M_t} d_i^{(t)} + U_t}{M_t + U_t}, \tag{8}$$

---

[8]This is a new, anonymous user.

where $d_i^{(t)}$ is the normalized distance of the $i$th matched pair and each unmatched box incurs a unit penalty. For a video with $T$ frames, the score is

$$\text{CM-Err} = \frac{1}{T}\sum_{t=1}^{T}\text{CM-Err}^{(t)}. \tag{9}$$

Lower values indicate better preservation of subject count and center locations, while higher values reflect additions or removals of subjects, merges or splits, and spatial displacements.

## C    EXPERIMENTS

Table 3: Comparison of different video editing methods on loveu-tgve-2023.

| Methods | Warp-Err ↓ | CLIP-T ↑ | CLIP-F ↑ | Q-Edit ↑ | CM-Err ↓ |
|---|---|---|---|---|---|
| VideoGrain (Yang et al., 2025) | 2.14 | 25.22 | 96.78 | 11.78 | 3.05 |
| TokenFlow (Geyer et al., 2023) | 2.23 | 24.22 | 97.20 | 10.86 | 3.54 |
| FateZero (Qi et al., 2023) | 2.18 | 23.80 | 97.11 | 10.91 | 3.75 |
| DMT (Yatim et al., 2024) | **1.90** | 23.82 | 97.18 | 12.53 | 3.61 |
| VideoPainter (Bian et al., 2025) | 2.12 | 22.95 | 95.95 | 10.82 | 4.04 |
| **IMAGEdit** | 2.04 | **25.99** | **97.23** | **12.74** | **2.66** |

**The loveu-tgve-2023 Dataset Results.** As noted above, we achieved strong results on the proposed MSVBench. To further validate our method, we also evaluate on the loveu-tgve-2023 dataset, where more than 80% of samples contain single or few subjects. As shown in Table 3, IMAGEdit attains the best semantic consistency and editing quality (CLIP-T 25.99, CLIP-F 97.23, Q-Edit 12.74) and the best layout and count preservation (lowest CM-Err 2.66), indicating better retention of category fidelity, subject centers, and counts after editing. For temporal and geometric stability, Warp-Err reaches 2.04, second only to DMT at 1.90, placing IMAGEdit in the leading group and balancing low distortion with high quality. Compared with VideoGrain and TokenFlow, IMAGEdit shows more balanced gains across metrics, demonstrating strong generalization and consistency in single or few subject scenarios.

**The Influence of $\tau$.** We vary the injection threshold $\tau$ from 0 to 50 to study how long the mask motion sequence should guide the denoising process. As shown in Figure 14, very small values of $\tau$ provide insufficient structural guidance, leading to boundary leakage, imperfect occlusion ordering, and occasional identity drift. In contrast, very large values inject fusion signals into late refinement steps and introduce artifacts such as texture corruption and visible seams. Mid range settings yield a better balance: around $\tau = 30$ the edits preserve structure and layering while allowing the backbone to synthesize high frequency details, producing clean boundaries and stable appearance. We therefore set $\tau = 30$ based on cross validation on a held out split.

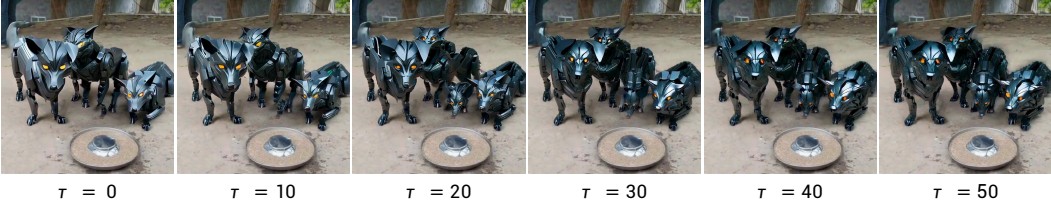

| $\tau = 0$ | $\tau = 10$ | $\tau = 20$ | $\tau = 30$ | $\tau = 40$ | $\tau = 50$ |

Figure 14: Ablation on $\tau$. The parameter $\tau$ is varied between 0 and 50 to systematically examine its effects. We show the last frame of the edit video

**More Qualitative Results.** Figure 15 presents additional side by side comparisons on continuous frames, covering fast motion, crowded scenes, and multi subject counts. Compared with baseline, IMAGEdit preserves category fidelity and identity consistency, produces cleaner boundaries, and yields edits with better temporal consistency, with fewer leakage artifacts and less flicker.

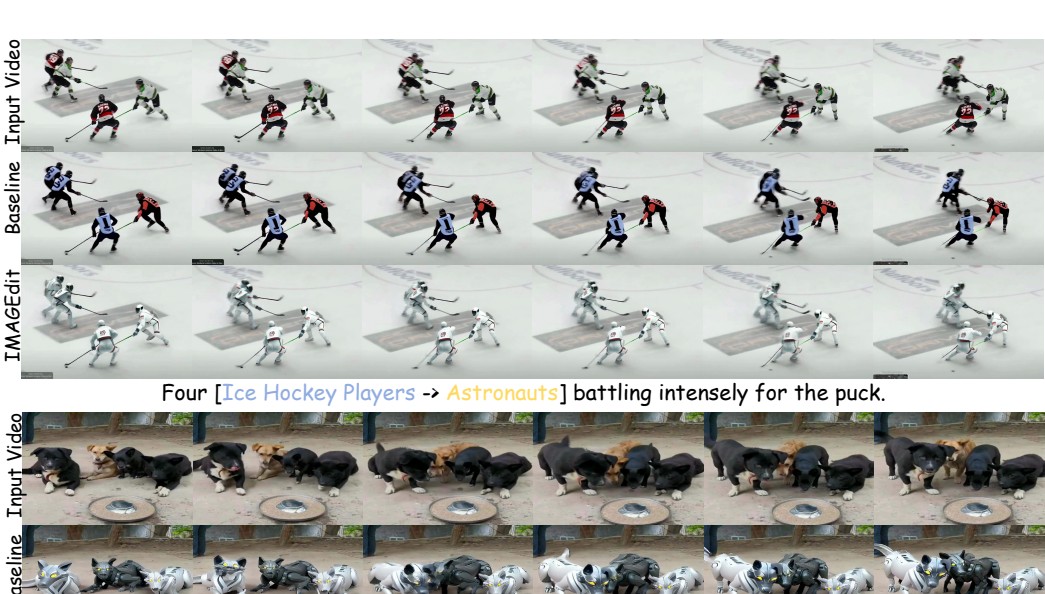

Four [Ice Hockey Players -> Astronauts] battling intensely for the puck.

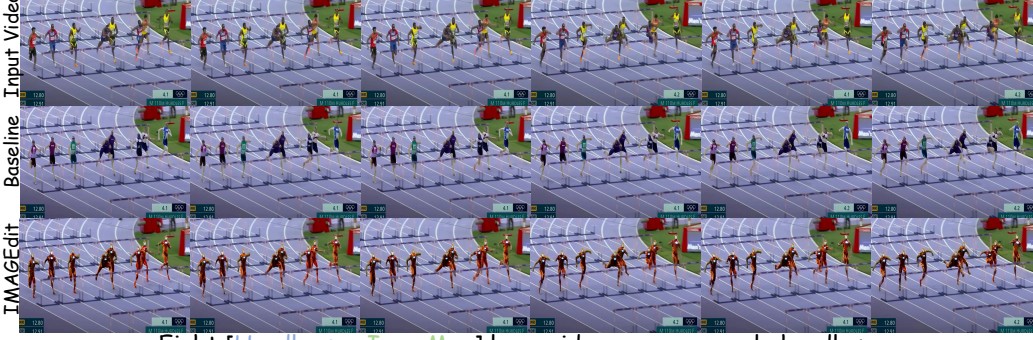

Four [Hungry Dogs -> Robot Wolves] surrounding a bowl of food outdoors.

Eight [Hurdlers -> Iron-Men] leap mid-race over purple hurdles.

Figure 15: More qualitative comparisons between IMAGEdit and baseline methods on the MSVBench dataset.

**More Applications Results.** Figure 16 showcases the extensible applications of IMAGEdit across diverse scenarios, including (a) background editing, (b) multi round editing, (c) specified subject editing, (d) long video editing, (e) face swapping, (f) partial editing, (g) clothing swapping, and (h) viewpoint change editing. These results indicate that IMAGEdit preserves non-target regions and maintains strong temporal consistency across tasks and complex scenes without fine-tuning.

## D    FUTURE WORK

Although IMAGEdit has demonstrated strong performance, future work can explore a parameterized motion and expression retargeting module built on latent diffusion representations. By driving subject level video editing with continuously controllable spatiotemporal parameters, we aim to further improve temporal consistency and editing accuracy in long horizon and heavy occlusion scenarios.

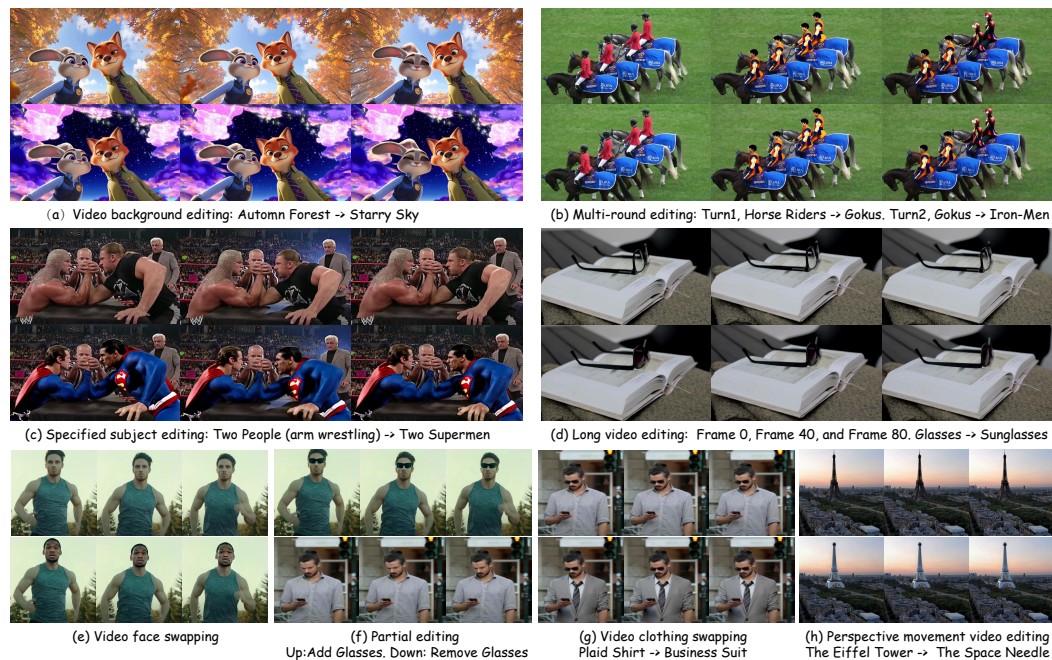

(a) Video background editing: Automn Forest -> Starry Sky

(b) Multi-round editing: Turn1, Horse Riders -> Gokus. Turn2, Gokus -> Iron-Men

(c) Specified subject editing: Two People (arm wrestling) -> Two Supermen

(d) Long video editing: Frame 0, Frame 40, and Frame 80. Glasses -> Sunglasses

(e) Video face swapping

(f) Partial editing
Up:Add Glasses. Down: Remove Glasses

(g) Video clothing swapping
Plaid Shirt -> Business Suit

(h) Perspective movement video editing
The Eiffel Tower -> The Space Needle

Figure 16: More qualitative comparisons on multi-scenario applications.

# E  LLM USAGE

Large language models (LLMs), specifically GPT-5, were used solely for grammar correction, sentence refinement, and to improve clarity and coherence in the Introduction and Method sections. All scientific content, experimental design, analyses, and conclusions were conceived, verified, and critically reviewed by the authors. The LLM did not generate factual claims, experimental results, or mathematical derivations.

