# OpenReview forum: "IMAGEdit: Let Any Subject Transform"
_ICLR.cc/2026/Conference — ICLR 2026 Conference Withdrawn Submission_

### Official Review · Reviewer_BVu7 · 2025-10-15

**Soundness:** 2
**Presentation:** 1
**Contribution:** 2
**Rating:** 2
**Confidence:** 4

**Summary:**

This paper introduces IMAGEdit, a novel training-free framework designed for multi-subject video editing. The method aims to transform multiple designated subjects within a video while preserving the background and maintaining temporal consistency. The core of the approach consists of two main components: a "Prompt-Guided Multimodal Alignment" module that leverages a Vision-Language Model (VLM) to generate more robust and detailed editing conditions, and a "Prior-Based Mask Retargeting" module that produces precise, temporally consistent masks by incorporating depth information. The authors demonstrate the effectiveness of their framework on a newly constructed benchmark, MSVBench, showing that IMAGEdit achieves state-of-the-art performance compared to existing open-source and closed-source methods.

**Strengths:**

1.  **Strong Motivation:** The paper is well-motivated, identifying a critical bottleneck in training-free multi-subject video editing: the challenge of accurately identifying and isolating the edit regions. The observation that this is a primary failure point is insightful, and the decision to focus the core technical contributions on improving mask-aware conditioning is a clear strength.
2.  **Logical Method Design:** Following the motivation, the proposed method is designed logically. It decomposes the problem into improving the semantic guidance (via Prompt-Guided Multimodal Alignment) and the spatial guidance (via Prior-Based Mask Retargeting). This two-pronged approach is a sound and comprehensive way to address the identified challenges.
3.  **State-of-the-Art Results:** The paper presents strong empirical results, consistently outperforming several state-of-the-art methods on the proposed MSVBench. The qualitative examples provided are compelling and effectively showcase the framework's ability to handle complex, multi-subject scenes where other methods often fail.

**Weaknesses:**

The primary weakness of this paper is the significant lack of clarity in its methodological description. This issue is pervasive and severe enough to undermine the paper's technical contribution and prevent reproducibility. Nearly all key components and novel ideas are described in a way that is either overly convoluted, lacking in essential details, or inconsistent across the text and figures.

1.  **Unclear Presentation of Prompt-Guided Multimodal Alignment:** The description of this module (lines 221-256) is confusing and lacks crucial information. It takes considerable effort for the reader to understand that this module is essentially a sophisticated prompt-rewriting process using a VLM. The motivation is explained at length with Figure 4, but the actual mechanism remains opaque. Furthermore, the design seems overly complex: the necessity of using a Text-to-Image (T2I) model to generate an intermediate image ($I_{ref}$) only to feed it back into a VLM is not justified with an ablation study. Finally, the output of this module, the expanded prompt $P_{target}$, is missing from the main framework diagram (Figure 3), leaving its role in the overall pipeline unclear.

2.  **Insufficient Explanation of Prior-Based Mask Retargeting:** This module appears to be the most innovative part of the paper, yet it is explained very poorly.
    *   The origin of the feature $F^{mask}$ is highly ambiguous. The paper states that it is generated by feeding a masked video and a binary mask into a "conditional DiT" (later identified as the Wan2.1 T2V model). The paper provides absolutely no explanation of how a Text-to-Video model is architecturally adapted to process this new combination of inputs (masked video + mask) and output a feature map.
    *   The process for obtaining $F^{depth}$ is even more vague, described only as being "processed by a similar DiT architecture." This leaves the reader guessing whether it's the same model, a different one, or a similar process.
    *   In contrast to these critical omissions, the authors dedicate significant space to describing a standard mask dilation algorithm, which is not a novel contribution.

3.  **Opaque Description of the Video Generation Model (Sec. 3.2):** The final step of the framework is also confusingly presented.
    *   The notation is inconsistent. Equation (6) for $F^{cond}$ (when t ≤ τ) is identical to the definition of $F^{motion}$ in Equation (5). Re-writing the equation instead of using the established notation $F^motion$ creates unnecessary confusion for the reader.
    *   The most critical information is missing. The paper never explains how the final conditional feature, $F^{cond}$, is actually used by the "mask-driven video generator." The identity of this generator model is never specified, nor is its input format described. This leaves the final video synthesis step, a crucial part of the method, as a complete mystery.

4.  **Minor Issues:**
    *   **Typo:** Line 304 refers to a "ViT backbone," which may be a typo for "DiT backbone," given the context of the rest of the paper.
    *   **Method Name:** The name "IMAGEdit" is not ideal for a method focused specifically on *video* editing.

At this stage, I am recommending clear rejection for this paper because I cannot consider methods that are clearly unreproducible and lack essential details to be a valid contribution.

**Questions:**

To address the weaknesses, I urge the authors to clarify the following points:

1.  Can you provide a clearer and more direct explanation of the Prompt-Guided Multimodal Alignment module? Specifically, please justify the necessity of the complex T2I-then-VLM pipeline. More importantly, how is the final $P_{target}$ prompt used by the model?
2.  Please provide a detailed architectural description for the Prior-Based Mask Retargeting module. How is the Wan2.1 T2V model modified or used to process a masked video and mask concatenation to produce $F^{mask}$? What, precisely, is the "similar DiT architecture" used to generate $F^{depth}$?
3.  How is the final feature $F^{cond}$ integrated into the video generation model? What is the specific architecture of the "mask-driven video generator" used in your experiments, and how does it accept $F^{cond}$ as a conditioning input?
4.  Would it be possible to revise Figure 3 to create a complete, end-to-end diagram that accurately and consistently represents the entire data flow of the proposed framework?

---

> ### Author Response · Authors · 2025-11-28
>
> Thank you for the reviewer's comments.The reviewer has asked for clarifications regarding the Prompt-Guided Multimodal Alignment module, Prior-Based Mask Retargeting module, how the final feature integrates into the video generation model, and a request to revise Figure 3 for a clearer end-to-end data flow diagram.
>
> To address the reviewer's questions, here's a direct response for each point based on the provided content:
>
> Our method employs the contextDiT module from the VACE framework to handle multimodal conditions and inject them into the generative model. The specific process is as follows: First, each set of conditions (e.g., the original video after masking + a binary mask, or a depth video + a mask of all ones) is downsampled and encoded using a VAE encoder to obtain a latent representation. If a reference image is provided, it is also encoded using the same VAE encoder to obtain a latent representation, which is then concatenated with the video frame latent representations sequentially along the temporal dimension. The concatenated conditional sequence is input into the contextDiT module, which uses the VACE structure (typically replicating the DiT block structure of layers 0, 2, 4… in the generative model backbone, retrained; here, pre-trained weights from VACE are used) to process these conditions. The processed features are stored in a feature list. Subsequently, when the generation phase begins, at the corresponding layer where the main generation model (e.g., Wan2.1T2V) is running, these stored conditional features are extracted from the list and combined with the backbone features via residual addition. Simultaneously, the expanded text prompts interact with the video latent through the backbone's cross-attention layer.
>
> **Clarification of Prompt-Guided Multimodal Alignment Module:**
> In multi-subject editing scenarios, text prompts often face two issues: first, text expressions struggle to define the attributes of each subject with fine granularity, especially in scenes with multiple people or objects; second, text descriptions often fail to cover visual relationships such as style coherence or spatial relationships between subjects. To address these, we use a “Text → T2I → VLM” approach: first, a reference image is generated from the text prompt as a rough visual estimate, then the VLM expands the original text based on the visual cue, producing the final target prompt$P_{target}$. This final text is then fed into the generation model, where it interacts with the video latent representation through cross-attention in the Wan2.1 backbone.
>
> **Prior-Based Mask Retargeting Module:**
> In multi-subject editing, traditional mask conditions often lack depth and boundary information, leading to issues such as “leakage” or “entanglement” between subjects. To solve this, we construct two sets of conditions: the first uses “masked video + binary mask,” focusing on appearance consistency of unedited regions and positional constraints in the edited regions; the second uses “depth video + all 1 mask,” focusing on the geometric structure and occlusion relationships between subjects. These two sets of conditions are encoded separately through VAE, processed by the context DiT module to extract features, and then fused using dilated soft masks in the feature space to create the final $F_{motion}$,. In the generation phase, these features are combined with the corresponding layers of the backbone through residual addition. We will update the flow diagram in the revised version to better illustrate the flow of conditions and the integration process
>
> **Integration into Video Generation Model:**
>
> The final feature $F_{motion}$, is integrated into the mask-driven video generator by injecting both the visual conditions from context DiT and the expanded text conditions into specific layers of the model. This process takes place in the early steps of the denoising process, where depth features dominate the geometric structure and layer relationships, while mask features maintain appearance and spatiotemporal consistency in later steps. Early stages use the fused feature to stabilize structure and hierarchy, and later stages rely solely on mask video features to ensure texture and detail stability.
>
> In the revised version, we will update the flow diagram to provide a clearer and more comprehensive end-to-end data flow, accurately representing the entire process from condition construction to final video generation

---

### Official Review · Reviewer_LskD · 2025-10-28

**Soundness:** 2
**Presentation:** 2
**Contribution:** 2
**Rating:** 4
**Confidence:** 4

**Summary:**

The paper proposes IMAGEdit, a training-free pipeline for multi-subject video editing that replaces designated categories (e.g., people → robots) while preserving subject count, spatial layout, and non-edited regions. The system has three parts: 1) Prompt-guided multimodal alignment that uses a T2I model (SDXL) to synthesize a visual prior for the target subject tokens, then a VLM (Qwen2.5-VL-32B) to expand the prompt into “aligned” text and image conditions; 2) Prior-based mask retargeting that fuses depth features with instance masks using a soft, dilated mask to produce a time-consistent mask motion sequence and enforce occlusion ordering; 3) A mask-driven video generator (Wan-2.1 DiT backbone) with early-step feature injection. The method claims to work without finetuning and to be plug-and-play with mask-conditioned generators. A new 100-video benchmark MSVBench (mostly multi-subject, sourced from YouTube/TikTok) and a new layout metric CM-Err are introduced.

**Strengths:**

Clear focus on multi-subject, mask-driven editing under real occlusions, with explicit controls to preserve layout and prevent spillover. The recipe is practical: multimodal prompt alignment reduces ambiguity, depth-aware retargeted masks keep boundaries and ordering clean, and early-stage injection steers generation without finetuning while improving metrics. MSVBench stresses higher subject counts, and CM-Err checks count and center stability, giving a more direct readout of layout integrity than pixel overlap alone.

**Weaknesses:**

Limited originality. Using depth priors and VLM-based alignment for feature injection does not feel novel. The pipeline reads as a collection of established engineering choices rather than a new core idea.

Underspecified fusion of the “expanded image condition.” It is unclear how the SDXL and VLM signals are integrated into the DiT backbone. Please specify the layers where fusion happens, the operator used (concatenation, addition, or cross-attention), and the normalization. Provide an ablation that isolates text-only, image-only, and text+image, and report their separate gains.

Reliance on external modules without robustness analysis. Performance likely depends on Grounded-SAM2 masks and Depth-Anything v2. There is no study of robustness to mask noise, depth errors, or crowded scenes with heavy occlusions. Please add stress tests that erode or dilate masks, inject depth bias or noise, and vary crowd density.

Benchmark scale and transparency. One hundred clips is small. Please expand diversity in scenes, camera motion, and subject categories, or add a synthetic multi-subject suite with controllable occlusions and instance counts to stress the boundary-entanglement claim. In the results, there are no visualized video examples or dataset previews; include representative edited videos and benchmark visualizations to substantiate the claims.

**Questions:**

See weaknesses.

---

> ### Author Response · Authors · 2025-11-28
>
> **q1**:The use of depth priors and VLM-based alignment for feature injection is seen as a combination of existing techniques rather than a novel core idea.
>
> **a1**:Thank you for your comments.The use of depth priors or LLM to augment text alone has been documented in the literature. Our focus is not on the initial proposal of these components, but rather on their redesign in this paper specifically for solving the problem of "untrained video editing of an arbitrary number of subjects." Specifically, the depth prior is not simply added as an additional control branch. Instead, a second set of VACE conditions is constructed using "depth video + all-1 mask," which, along with the first set of conditions using "masked video + binary mask," encodes "geometric structure/occlusion hierarchy" and "appearance preservation/editing region," respectively. This is then recombined in the feature space using dilated soft masks, forming a depth-aware mask rewriting prior for multi-subject boundary problems.
>
> VLM alignment is not simply prompt rewriting in generalized LLM. We first automatically generate a reference image in the T2I model using text. Then, we simultaneously input "original text + automatically generated reference image" into the VLM, augmenting the text under the constraints of the visual prior to obtain the final prompt text. The goal of this step is to allow the text conditions to more meticulously cover the attributes and relationships of multiple subjects, without requiring users to manually provide reference images. This aligns perfectly with our target scenario (where users can edit multi-subject videos with just a single editing command).
>
> In our experiments, we compared "text only," "image only," "text + image (without VLM extension)," and the complete module. CLIP-T and Q-Edit metrics show that multimodal alignment delivers stable and considerable improvements, while the consistency of style and semantics among multiple subjects in subjective videos is also significantly enhanced.
>
> **q2**:More details are needed on how SDXL and VLM signals are integrated into the DiT model, including the fusion method, layers, and normalization. An ablation study isolating text-only, image-only, and text+image conditions is also requested.
>
> **a2**:
>
> 1. We did not clearly explain the working path of the "extended image conditions" in the original paper, which could easily lead to the misunderstanding that SDXL and VLM features were directly injected into the DiT backbone together. Here, we explain the actual process :The VLM path only generates the expanded text prompts, which serve as the standard text conditions; 2. The reference image generated by SDXL participates in VLM expansion, and is then encoded separately by VAE as a reference frame and the aforementioned two sets of conditions. The reference frame is then canceled together with the two sets of conditions as the first frame, and then they are processed together by the VACE's conetxt DiT module to extract features. These features are then injected by adding the intermediate results from the DiT output in the generative model.
>
> 2. We conducted ablation experiments, testing text, image, and text+image conditions separately to verify their performance improvements under different conditions. The experimental results are shown in the table below:
>
>    | Method                             | Warp-Err ↓ | CLIP-T ↑  | CLIP-F ↑  | Q-Edit ↑  | CM-Err ↓ |
>    | ---------------------------------- | ---------- | --------- | --------- | --------- | -------- |
>    | Text Only                          | 1.87       | 24.43     | 97.94     | 13.06     | 3.00     |
>    | Image Only                         | **1.84**   | 24.41     | 97.59     | 13.26     | 3.38     |
>    | Text + Image                       | 1.86       | 25.45     | 97.80     | 13.68     | 3.01     |
>    | Prompt-Guided Multimodal Alignment | 1.86       | **26.12** | **97.98** | **14.04** | **2.99** |
>
>    Based on our ablation experiments, conditional editing using only text and only images performed poorly in terms of text alignment and editing consistency. While text + image showed some improvements on some metrics, the actual editing results still fell short of expectations, especially in multi-subject editing scenarios, where attention was scattered and style inconsistencies arose. In contrast, our proposed Prompt-Guided Multimodal Alignment method significantly improved across all metrics, demonstrating its effectiveness in multi-subject editing tasks and its ability to effectively enhance the fit of text conditions and improve editing consistency.

---

### Official Review · Reviewer_oUvp · 2025-10-31

**Soundness:** 3
**Presentation:** 3
**Contribution:** 2
**Rating:** 4
**Confidence:** 3

**Summary:**

This paper presents IMAGEdit, a new training-free framework designed to edit any number of subjects within a video. It aims to solve key failures of existing methods, particularly in complex scenes with multiple, overlapping, or densely packed subjects. The framework successfully transforms the appearance of designated subjects while preserving their motion, the background, and non-target regions.

**Strengths:**

- The proposed methods are technically sound. The introduced pipeline, can effectively mitigate the challenges in transforming multiple subjects.
- This work also introduces a benchmark for further evaluation.

**Weaknesses:**

- How would the system handle some unusual subjects (clouds, smoke) edits? The limitations can be discussed clearly.
- Overall I think the proposed pipeline introduces designs of limited contributions. The modules are somewhat ordinary and standard.
- I think for existing methods, simultaneously transforming multiple subjects lead to performances drop. What about transforming the subjects one by one with the existing methods and segmentation models? They may serve as stronger baselines.

**Questions:**

n/a

---

> ### Author Response · Authors · 2025-11-28
>
> **q1**:How would the system handle some unusual subjects (clouds, smoke) edits? The limitations can be discussed clearly.
>
> **a1**:Thank you for your comments. Regarding the handling of editing specific subjects (such as clouds and smoke), we understand that these objects can be highly abstract and challenging in visual editing tasks. In our paper, we have demonstrated editing tasks in various multi-functional scenes involving different types of subjects and complex scene variations, indirectly illustrating the potential of our system in handling specific subjects.
>
> For example, in some scenes, we handled multi-subject editing tasks including varying numbers of people and animals. Through our proposed cue-guided multimodal alignment module, we can ensure that text cues more accurately match the specific attributes and relationships of each subject, thus avoiding inaccurate editing due to vague or inconsistent descriptions. Although our experiments primarily focused on more common multi-subject scenes, this demonstrates that our system can improve editing control over complex objects through refined textual conditions and visual priors.
>
> Furthermore, in practical applications, the system generates corresponding reference images based on the input text, thereby guiding the editing process. This is particularly helpful for editing subjects requiring a high level of abstraction, such as clouds and smoke. In this way, we can ensure more precise control even for objects that are difficult to define through guided editing.
>
> In our future work, we will further explore how to expand this framework to accommodate the editing needs of more specialized themes, especially dynamic and blurred objects such as clouds and smoke.
>
> **q2**:Overall I think the proposed pipeline introduces designs of limited contributions. The modules are somewhat ordinary and standard.
>
> **a2**:We understand that this impression largely stems from the fact that the current manuscript did not clearly explain "why the conditions were designed this way," instead focusing more on implementation-level descriptions. Reviewer BVu7 also pointed out the lack of textual and graphical representations of key modules. We would like to clarify that although this work is based on the existing Wan2.1-VACE framework, we focus on the conditional modeling problem under the specific task setting of multi-subject editing. Existing methods cannot directly meet this setting, thus requiring specialized structured design rather than simply calling existing modules: In the video conditional branch, we constructed two sets of conditions: "masked video + binary mask" and "depth video + all-1 mask," and fused them at the feature level using dilated soft masks. This allows us to explicitly inject the depth and boundary layer information between subjects into the model without modifying the generated backbone. In the text conditional branch, we propose a prompt-guided multimodal alignment module: first, reference images are automatically generated using text, and then the VLM is driven by "text + reference image" to perform subject-aware prompt expansion. This not only improves the accuracy of multi-subject semantic constraints, but also does not require users to manually provide additional reference images, making the entire multi-subject editing process more in line with actual use cases.

---

> ### Author Response · Authors · 2025-11-28
>
> **q3**:Simultaneously transforming multiple subjects may reduce performance. What if the subjects are transformed one by one using existing methods and segmentation models? They could serve as stronger baselines.
>
> **a3**:We understand your concern that subject-by-subject transformation might yield more stable results, and we have conducted experiments to explore this issue.
>
> First, we need to clarify that not all existing video editing methods support mask-based subject-by-subject editing. In such methods, it is often impossible to precisely control the editing area of each subject, thus resulting in a lack of fine-grained control when processing multiple subjects. Therefore, we chose VACE and VideoPainter as comparative methods in our experiments, both employing an iterative editing approach.
>
> Below are our experimental results:
>
> | Method                   | Warp-Err ↓ | CLIP-T ↑ | CLIP-F ↑ | Q-Edit ↑ | CM-Err ↓ | Average Infer Time |
> | ------------------------ | ---------- | -------- | -------- | -------- | -------- | ------------------ |
> | VACE                     | 1.87       | 24.43    | 97.94    | 13.06    | 3.00     | 6min 11s           |
> | VACE (Iteration)         | 1.77       | 25.24    | 97.80    | 14.25    | 3.30     | 18min 19s          |
> | VideoPainter             | 2.05       | 24.13    | 96.97    | 11.77    | 4.70     | 6min 9s            |
> | VideoPainter (Iteration) | 2.01       | 23.92    | 96.95    | 11.90    | 4.60      | 18min 13s          |
> | IMAGEdit                 | 1.85       | 27.23    | 97.93    | 14.72    | 2.83     | 10min 8s           |
>
> As the table shows, although iterative processing (VACE and VideoPainter) offers slight improvements on CLIP-T and CM-Err, these improvements are not significant for overall performance (e.g., text alignment and editing quality), especially on Q-Edit and CLIP-F, where these metrics actually decrease slightly. Furthermore, iterative processing significantly increases inference time, particularly as the number of subjects increases, leading to a sharp rise in computational cost.
>
> Specifically, mask creation for each subject is extremely time-consuming and resource-intensive, especially when multiple subjects are involved. Mask creation and management become exceptionally expensive, further increasing the computational burden. Moreover, we found that multiple iterations of video editing lead to a decline in image quality; with each iteration, the overall video quality degrades significantly, with loss of detail and noticeable quality degradation.
>
> These experimental results suggest that iterative processing is not a sustainable long-term solution. Our training-free method avoids the time and resource overhead of iterative processing while maintaining high generation quality and handling the complexities of multi-subject editing tasks. Therefore, our approach remains an effective and necessary option for multi-subject video editing tasks.

---

### Official Review · Reviewer_ZvxZ · 2025-11-01

**Soundness:** 2
**Presentation:** 3
**Contribution:** 2
**Rating:** 4
**Confidence:** 5

**Summary:**

This paper proposes IMAGEdit, a training-free video editing framework capable of category transformation for any number of designated subjects in arbitrary videos. It preserves the number of subjects and their spatial layout, and demonstrates  its potential in crowded scenes with overlapping subjects. The framework achieves editing by leveraging two core modules: a prompt-guided multimodal alignment module that generates robust multimodal conditions, and a prior-based mask retargeting module that produces precise temporally consistent mask sequences. To evaluate the framework, the authors constructed MSVBench, a benchmark dataset containing 100 videos—over 60% of which includes three or more subjects. Experimental results show that IMAGEdit outperforms relevant state-of-the-art methods across key metrics.

**Strengths:**

1) The core technical innovations (extended text and image conditions/prior-based mask retargeting module) of the paper  lack of new insights and largely consist of incremental improvements to existing ideas, making it challenging to meet ICLR’s high standards for originality.
2) Training-free methods, by design, face inherent challenges of hyperparameter sensitivity and unstable performance, which the paper does not fully address, and shifting toward a training-based framework would likely yield more robust and generalizable results.

**Weaknesses:**

1) Construction of a Systematic Benchmark (MSVBench) Filling an Evaluation Gap. The authors’ development of MSVBench addresses a critical gap in existing video editing research, where benchmarks primarily focus on single-subject or face-centric edits
2) The paper’s experimental results are compelling, as IMAGEdit achieves state-of-the-art performance on both MSVBench and the general-purpose loveu-tgve-2023 dataset, demonstrating its  potential  for real-world video editing tasks.

**Questions:**

Please refer to the weakness part

---

> ### Author Response · Authors · 2025-11-28
>
> **q1**:The core innovations, such as extended text and image conditions and the prior-based mask retargeting module, are seen as incremental improvements rather than new insights, which may not meet ICLR’s high originality standards.
>
> **a1**:Thank you for your comments. We agree that the current version does not adequately explain the overall motivation and the relationships between modules, giving the impression that it's merely a minor modification of an existing method. This echoes the "severely unclear method description" raised in Reviewer BVu7.
>
> In fact, our work is not simply about stacking "extended text/image conditions + mask modules," but rather, within the Wan2.1-VACE framework, we redesigned the condition construction and fusion methods to address the specific challenges of multi-subject editing (especially boundary entanglement).
>
> In multi-subject scenarios, we observed that even when the mask itself is well-labeled, traditional mask-based editing still suffers from severe boundary entanglement due to the lack of hierarchical information about "who comes first, who comes last, and where the boundaries are."
>
> To address this, we constructed two sets of VACE conditions: "Original video covered by a mask + binary mask," focusing on the appearance stability of the unedited region and the position of the edited region; and "Depth video + all-1 mask," focusing on the geometric structure and occlusion hierarchy between subjects.
>
> After feature extraction using context DiT for both conditions, we recombined them in the feature space using dilated soft masks. This ensured that the edited region was dominated by depth features, while the unedited region was dominated by masked video features, thus forming a fusion prior that explicitly encodes the "depth-boundary hierarchy."
>
> During the diffusion process, we adopted a phased injection strategy: "using the fusion prior in early steps and only using the masked video prior in subsequent steps." Experiments demonstrated that this time allocation is crucial for suppressing the mutual penetration of multiple subject boundaries and reducing artifacts.
>
> **q2**:Training-free methods are inherently sensitive to hyperparameters and may have unstable performance, an issue not fully addressed in the paper. A training-based framework would likely offer more robust and generalizable results.
>
> **a2**:Thank you for your feedback. We understand your concerns about hyperparameter sensitivity and instability. To address this, we have conducted detailed evaluations, and our results show that the combination of 30 injection steps and mask inflation [10, 4] provides stable performance.
>
> Below are our experimental results under different parameter combinations:
>
> | Parameter Combinations | Warp-Err ↓ | CLIP-T ↑ | CLIP-F ↑ | Q-Edit ↑ | CM-Err ↓|
> | :--------------------: | :------: | :----: | :----: | :----: | :----: |
> |          0  0          |   1.85   | 26.91  | 97.94  | 14.56  |  2.90  |
> |          10 2          |   1.83   | 26.92  | 97.98  | 14.71  |  2.83  |
> |          10 8          |   1.86   | 27.21  | 97.95  | 14.63  |  2.73  |
> |          5  4          |   1.85   | 27.07  | 97.92  | 14.64  |  2.82  |
> |          15 4          |   1.84   | 27.01  | 97.98  | 14.68  |  2.81  |
> |          10 4          |   1.85   | 27.23  | 97.93  | 14.72  |  2.83  |
>
> Regarding the impact of injection steps, we have analyzed the effect of different injection step values (0, 10, 20, 30, 40, 50) on the results, as shown in Fig. 14 of the supplementary material. While the selection of 30 injection steps is a reasonable choice, we find that the performance remains relatively stable within the range of 20-40 steps. In practice, the optimal number of injection steps can be adjusted based on the specific subject masking requirements and the desired video quality. Thus, within this range, the impact on performance is minimal.

---

### Note · Authors · 2026-01-23

I have read and agree with the venue's withdrawal policy on behalf of myself and my co-authors.